# Effects of common mutations in the SARS-CoV-2 Spike RBD and its ligand, the human ACE2 receptor on binding affinity and kinetics

Michael I Barton[1†], Stuart A MacGowan[2†], Mikhail A Kutuzov[1], Omer Dushek[1], Geoffrey John Barton[2]*, P Anton van der Merwe[1]*

[1]Sir William Dunn School of Pathology, University of Oxford, Oxford, United Kingdom; [2]School of Life Sciences, University of Dundee, Dundee, United Kingdom

**Abstract** The interaction between the SARS-CoV-2 virus Spike protein receptor binding domain (RBD) and the ACE2 cell surface protein is required for viral infection of cells. Mutations in the RBD are present in SARS-CoV-2 variants of concern that have emerged independently worldwide. For example, the B.1.1.7 lineage has a mutation (N501Y) in its Spike RBD that enhances binding to ACE2. There are also ACE2 alleles in humans with mutations in the RBD binding site. Here we perform a detailed affinity and kinetics analysis of the effect of five common RBD mutations (K417N, K417T, N501Y, E484K, and S477N) and two common ACE2 mutations (S19P and K26R) on the RBD/ACE2 interaction. We analysed the effects of individual RBD mutations and combinations found in new SARS-CoV-2 Alpha (B.1.1.7), Beta (B.1.351), and Gamma (P1) variants. Most of these mutations increased the affinity of the RBD/ACE2 interaction. The exceptions were mutations K417N/T, which decreased the affinity. Taken together with other studies, our results suggest that the N501Y and S477N mutations enhance transmission primarily by enhancing binding, the K417N/T mutations facilitate immune escape, and the E484K mutation enhances binding and immune escape.

*For correspondence:
g.j.barton@dundee.ac.uk (GJohnB);
anton.vandermerwe@path.ox.ac.uk (PAdM)

†These authors contributed equally to this work

## Introduction

Since its identification in 2019, a coronavirus able to induce a severe acute respiratory syndrome in humans, SARS-CoV-2, has resulted in arguably the most severe infectious disease pandemic in 100 years. To date, more than 135 million people have been infected, resulting in the deaths from the resulting disease, COVID-19, of more than 3 million people (**WHO, 2021**), and measures introduced to control spread have had harmful social and economic impacts. Fortunately, effective vaccines have been developed, and a global vaccination programme is underway (**Mahase, 2021**). New SARS-CoV-2 variants of concern are emerging that are making containment of the pandemic more difficult, perhaps by increasing transmissibility of the virus (**Davies et al., 2021**; **Korber et al., 2020**; **Volz et al., 2021b**; **Volz et al., 2021b**; **Washington et al., 2021**) and/or its resistance to protective immunity induced by previous infection or vaccines (**Darby and Hiscox, 2021**; **Dejnirattisai et al., 2021**; **Garcia-Beltran et al., 2021**; **Madhi et al., 2021a**; **Madhi et al., 2021b**; **Mahase, 2021**; **Volz et al., 2021b**; **Volz et al., 2021b**).

The SARS-CoV-2 virus enters cells following an interaction between the Spike (S) protein on its surface with angiotensin-converting enzyme 2 (ACE2) on cell surfaces (**V'kovski et al., 2021**). The receptor-binding domain (RBD) of the Spike protein binds the membrane-distal portion of the ACE2 protein. The S protein forms a homotrimer, which is cleaved shortly after synthesis into two fragments that remain associated non-covalently: S1, which contains the RBD, and S2, which mediates membrane

**eLife digest** As the COVID-19 pandemic has progressed, new variants of the virus SARS-CoV-2 have emerged that are more infectious than the original form. The variants known as Alpha, Beta and Gamma have mutations in a protein on the virus's surface that is vital for attaching to cells and infecting them. This protein, called Spike, carries out its role by binding to ACE2, a protein on the surface of human cells. Mutations on Spike are found on the region where it binds to ACE2.

The interaction between these two proteins appears to be important to the behaviour of SARS-CoV-2, but the impact of individual mutations in Spike is unknown. In addition, some people have different variants of ACE2 with mutations in the region that interacts with Spike, but it is not known whether this affects these people's risk of contracting COVID-19.

To answer these questions, Barton et al. measured the precise effect of mutations in Spike and ACE2 on the strength of the interaction between the two proteins. The experiments showed that three of the five common Spike mutations in the Alpha, Beta and Gamma SARS-CoV-2 variants strengthened binding to ACE2. The two mutations that weakened binding were only found together with other mutations that strengthened binding. This meant that the Spike proteins in all three of these SARS-CoV-2 variants bind to ACE2 more strongly than the original form. The experiments also showed that two common variants of ACE2 also increased the strength of binding to Spike. Interestingly, one of these ACE2 variants reversed the effect of a specific SARS-CoV-2 mutation, suggesting that carriers would be resistant to SARS-CoV-2 variants with this mutation.

Identifying the precise effects of Spike mutations on ACE2 binding helps understand why new variants of SARS-CoV-2 spread more rapidly. This could help to identify concerning new variants before they spread widely and inform the response by health authorities. The finding that two common ACE2 variants bind more strongly to Spike suggests that people with these mutations could be more susceptible to SARS-CoV-2.

fusion following the binding of Spike to ACE2 (*V'kovski et al., 2021*). During the pandemic, mutations have appeared in the Spike protein that may increase transmissibility (*Davies et al., 2021*; *Korber et al., 2020*; *Richard et al., 2021*; *Volz et al., 2021b*; *Volz et al., 2021b*; *Washington et al., 2021*). One that emerged early in Europe, D614G, and quickly became dominant globally (*Korber et al., 2020*), increases the density of intact Spike trimer on the virus surface by preventing premature dissociation of S1 from S2 following cleavage (*Zhang et al., 2021*; *Zhang et al., 2020*). A later mutation, N501Y, which has appeared in multiple lineages, lies within the RBD, and increases its affinity for ACE2 (*Starr et al., 2020*; *Supasa et al., 2021*). These findings suggest that mutations that directly or indirectly enhance Spike binding to ACE2 may increase transmissibility.

Prior infection by SARS-CoV-2 and current vaccines induce antibody responses to the Spike protein, and most neutralising antibodies appear to bind to the Spike RBD (*Garcia-Beltran et al., 2021*; *Greaney et al., 2021a*; *Rogers et al., 2020*). Some variants of concern have mutations in their RBD that confer resistance to neutralising antibodies (*Darby and Hiscox, 2021*; *Dejnirattisai et al., 2021*; *Garcia-Beltran et al., 2021*; *Madhi et al., 2021a*; *Madhi et al., 2021b*; *Mahase, 2021*). What is less clear is the precise effect of these mutations on the affinity and kinetics of the binding of RBD to ACE2. Previous studies of the interaction between the Spike RBD and ACE2 have produced a wide range of affinity and kinetic estimates under conditions (e.g. temperature) that are not always well defined (*Lei et al., 2020*; *Shang et al., 2020*; *Supasa et al., 2021*; *Wrapp et al., 2020*; *Zhang et al., 2021*; *Zhang et al., 2020*). Precise information is needed to assess the extent to which RBD mutations have been selected because they enhance ACE2 binding or facilitate immune evasion.

In this study, we undertook a detailed affinity and kinetic analysis of the interaction between Spike RBD and ACE2 at a physiological temperature (37 °C), taking care to avoid common pitfalls. We used this optimised approach to analyse the effect of important common mutations identified in variants of RBD and ACE2. Both mutations of ACE2 (S19P, K26R) and most of the mutations of RBD (N501Y, E484K, and S477N) enhanced the interaction, with one RBD mutation (N501Y) increasing the affinity by ~10 -fold. Increased binding was the result of decreases in dissociation rate constants (N501Y, S477N) and/or increases in association rate constants (N501Y, E484K). Although the K417N/T mutations found in the South African (B.1.351) and Brazilian (P.1) variants both decreased the affinity,

the affinity-enhancing N501Y and E484K mutations that are also present in both variants confer a net ~4 -fold increase in the affinity of their RBDs for ACE2.

## Results
### Selection of variants

The focus of this study was to analyse common and therefore important variants of RBD and ACE2. Henceforth, we will refer to the common ACE2 allele and RBD of the original SARS-CoV-2 strain sequenced in Wuhan as wild type (WT). We chose mutations of RBD within the ACE2 binding site that have appeared independently in multiple SARS-CoV-2 lineages/clades (*Figure 1*, *Figure 1—figure supplement 1*; *Hodcroft, 2021*; *Rambaut et al., 2020*), suggesting that they confer a selective advantage, rather than emerged by chance, such as through a founder effect. The N501Y mutation has appeared in the Alpha (B.1.1.7; 20I/501Y.V1), Beta (B.1.351; 20 H/501Y.V2), and Gamma (P.1; 20 J/501Y.V3) variants, which were first identified in the UK, South Africa, and Brazil, respectively. The E484K mutation is present in the Beta and Gamma variants and has appeared independently in many other lineages, including Zeta (P.2; 20B/S.484K), B.1.1.318, Eta (B.1.525; 20A/S:484 K), and Iota (B.1.526; 20 C/S.484K). E484K has also appeared in VOC-202102–02, a subset of the Alpha variant identified in the UK (*SARS-CoV-2 Variants of concern and variants under investigation - GOV.UK, 2021*). The S477N mutation became dominant for periods in Australia (clade 20 F) and parts of Europe (20A.EU2) and then appeared in New York in the Iota or B.1.526 lineage (*Zhou et al., 2021a*). Mutations of K417 have appeared independently in the Beta and Gamma variants. Interestingly, N501Y, E484K, and S477N were the main mutations that appeared following random RBD mutagenesis and in vitro selection of mutants with enhanced ACE2 binding (*Zahradník et al., 2021*).

We selected for analysis the two most common mutations of ACE2 within the RBD binding site, K26R and S19P (*Figure 1C*). They are present in 0.4% and 0.03%, respectively, of all samples in the gnomAD database (*Karczewski et al., 2020*), while other ACE2 mutations in the RBD binding site are much less frequent (<0.004%) (*MacGowan et al., 2021*). K26R is observed in all the major gnomAD populations but is most common in Ashkenazi Jews (1%) and (non-Finnish) north-western Europeans (0.6%). It is less common in Africans/African-Americans and South Asians (0.1%) and rare in Finnish (0.05%) and East-Asian (0.001%) populations. The S19P mutant is almost exclusively found in Africans/African-Americans (0.3%).

### Measurement of affinity and kinetics

To measure the effects of these mutations on the affinity and kinetics of the RBD/ACE2 interaction, we used surface plasmon resonance (SPR), which allows very accurate measurements, provided that common pitfalls are avoided, particularly protein aggregation, mass-transport limitations, and rebinding (*van der Merwe and Barclay, 1996*; *Myszka, 1997*). Monomeric, soluble forms of the ecto-domain of ACE2 and the Spike RBD were expressed in human cells, to retain native glycosylation, and purified (*Figure 2—figure supplement 1*). ACE2 was captured onto the sensor surface via a carboxy-terminal biotin and RBD injected over ACE2 at different concentrations (*Figure 2A*). Excellent fits of 1:1 Langmuir binding model to the data yielded an association rate constant ($k_{on}$) of 0.9 ± 0.05 $\mu M^{-1}s^{-1}$ and a dissociation rate constant ($k_{off}$) of 0.067 ± 0.0011 $s^{-1}$ (mean ± SD, n = 6, *Table 1*). These rate constants are up to 25- fold faster than previously reported for the same interaction (*Lei et al., 2020*; *Shang et al., 2020*; *Supasa et al., 2021*; *Wrapp et al., 2020*; *Zhang et al., 2021*). However, previous experiments were conducted at unphysiologically low temperatures (i.e. below 37 °C) and under conditions in which mass-transport limitations and rebinding are highly likely (see Discussion). These factors, and the presence of protein aggregates (*van der Merwe and Barclay, 1996*), would all lower the measured rate constants. In contrast, our measurements were conducted at 37 °C and under conditions in which mass-transfer limitation and rebinding were excluded. The latter is demonstrated by the fact that measured $k_{on}$ and $k_{off}$ rates approached maximal values at the low level of ACE2 immobilisation (~50 RU) used in our experiments (*Figure 2B,C*; *Dejnirattisai et al., 2021*). The excellent fit of the 1:1 binding model to our data excludes an effect of protein aggregates, which yield complex kinetics. The calculated dissociation constant ($K_D$) was 74 ± 4 nM (mean ± SD, n = 6, *Table 1*). We also measured $K_D$ by equilibrium binding (*Figure 2D*), which avoids any artefacts induced by mass transfer limitations and rebinding. This $K_D$ was very similar to the value calculated from kinetic data

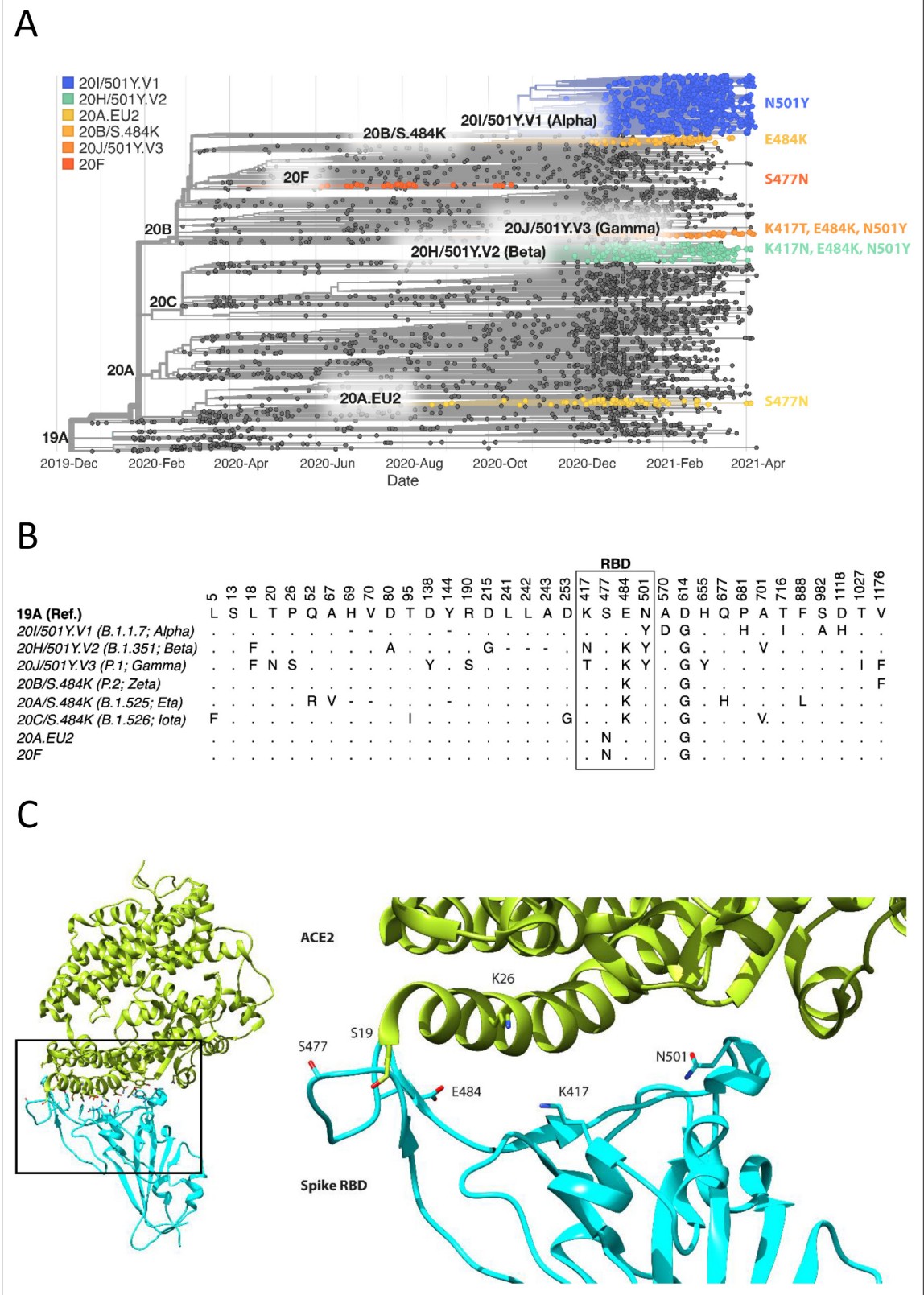

**Figure 1.** Spike RBD and ACE2 variants analysed in this study. (**A**) Phylogenetic tree illustrating the clades containing the RBD mutations investigated in this study. Constructed using TreeTime (*Sagulenko et al., 2018*) from the Nextstrain Global (*Hadfield et al., 2018*) sample of SARS-CoV-2 sequences from the GISAID database (*Shu and McCauley, 2017*) (accessed 15 April 2021, N = 4017). (**B**) Alignment illustrating the Spike residues that differ between SARS-CoV-2 variants, with the RBD mutants boxed. The variants are labelled with their clade designation from Nextstrain (*Hadfield et al.,*

*Figure 1 continued on next page*

*Figure 1 continued*

*2018*) and/or PANGO lineage (*Rambaut et al., 2020*), where relevant. The RBD mutations were collated from CoVariants (*Hodcroft, 2021*) and Nextstrain. (**C**) The structure of human ACE2 (green) in complex with SARS-CoV-2 Spike RBD (cyan). The area enclosed by the box is shown enlarged on the right, with the residues mutated in this study labelled. Drawn using UCSF Chimera (*Pettersen et al., 2004*) using coordinates from PDB 6m0j (*Lan et al., 2020*).

The online version of this article includes the following figure supplement(s) for figure 1:

**Figure supplement 1.** Emergence of the same RBD mutations in multiple SAR2-CoV-2 clades.

(63 ± 7.7 nM [mean ± SD, n = 24, *Table 1*]), and did not vary with immobilisation level (*Figure 2E*), further validating our kinetic measurements. These affinity values are within the wide range reported in previous studies, which varied from $K_D$ 6–133 nM (*Laffeber et al., 2021*; *Lei et al., 2020*; *Liu et al., 2021*; *Shang et al., 2020*; *Supasa et al., 2021*; *Wrapp et al., 2020*; *Zhang et al., 2021*).

## The effect of RBD mutations

We next evaluated the effect of RBD mutations on the affinity and kinetics of binding to ACE2 (*Figure 3* and *Table 1*). Example sensorgrams are shown of mutations that increased (N501Y, *Figure 3A*) or decreased (K417N, *Figure 3B*) the binding affinity, while the key results from all mutants are summarised in *Figure 3C*. The single mutations S477N, E484K, and N501Y all enhanced binding. The N501Y mutation had the biggest effect, increasing the affinity ~10 fold to $K_D$ ~7 nM, by increasing the $k_{on}$ ~1.8 -fold and decreasing the $k_{off}$ by ~7 -fold. The S477N and E484K mutations increased the affinity more modestly (~1.5 -fold), by decreasing the $k_{off}$ (S477N) or increasing the $k_{on}$ (E484K). The K417T and K417N mutations decreased the affinity ~2- and ~ 4 -fold, respectively, mainly by decreasing the $k_{on}$ but also by increasing the $k_{off}$. Affinity-altering mutations in binding sites mainly affect the $k_{off}$ (*Agius et al., 2013*) and have more modest effects on the $k_{on}$. Changes in electrostatic interactions can dramatically affect the $k_{on}$ (*Schreiber and Fersht, 1996*) and are a plausible explanation for the effects of the mutations K417T, K417N, and E484K on $k_{on}$. K417 forms a salt bridge with D30 on ACE2 (*Lan et al., 2020*), while E484 is ~9 Å from E75 on ACE2 (*Lan et al., 2020*). Thus, the mutations K417N/T and E484K would decrease and increase, respectively, long-range electrostatic forces that may accelerate association (*Schreiber and Fersht, 1996*).

We also examined the effect on ACE2 binding of combinations of RBD mutations, including combinations present in VOC-202102–02, a subset of the Alpha lineage (N501Y) with the E484K mutation ("SARS-CoV-2 Variants of concern and variants under investigation – GOV.UK," 2021), and the Beta and Gamma variants (*Figure 3C*, *Table 1*). In the case of VOC-202102–02, the addition of the E484K mutation to N501Y further increased the affinity, to ~15 -fold higher than WT RBD ($K_D$ ~5 nM), by further increasing the $k_{on}$. Because the higher $k_{on}$ could result in mass transfer limiting binding, we confirmed that the kinetic measurement for this variant was not substantially affected by varying levels of immobilisation (*Figure 3—figure supplement 1*). The affinity of the Beta (K417N/ E484K/N501Y) and Gamma (K417T/E484K/N501Y) RBD variants for ACE2 increased by 3.7- and 5.3-fold, respectively, relative to wild-type RBD, by both increasing the $k_{on}$ and decreasing the $k_{off}$ rate constants.

We next examined whether the effects of the mutations were additive, as is typically the case for multiple mutations at protein/protein interfaces (*Wells, 1990*). To do this, we converted the changes in $K_D$ to changes in binding energy ($\Delta\Delta G$, *Table 2*) and examined whether the $\Delta\Delta G$ measured for RBD variants with multiple mutations was equal to the sum of the $\Delta\Delta G$ values measured for the individual RBD mutants. This was indeed the case (*Figure 3D*), indicating that the effects on each mutation are independent. This is consistent with them being spaced well apart within the interface (*Figure 1C*) and validates the accuracy of the affinity measurements.

## The effects of ACE2 mutations

We next examined the effects of mutations of ACE2 (S19P and K26R) on binding to both wild-type and common variants of RBD (*Figure 4*, *Figure 4—figure supplement 1*, and *Table 1*). Both S19P and K26R increased the affinity of WT RBD binding by ~3.7- and ~ 2.4 -fold (*Figure 4A*). These increases in affinity were the result of both increases in the $k_{on}$ and decreases in the $k_{off}$.

Finally, we looked for interactions between RBD and ACE2 mutations by measuring the effects of the ACE2 mutations on binding to all mutant forms of RBD (*Table 1*). After converting changes in $K_D$ to $\Delta\Delta G$ (*Table 2*), we examined whether $\Delta\Delta G$ measured for a given ACE2 variant/RBD variant interaction

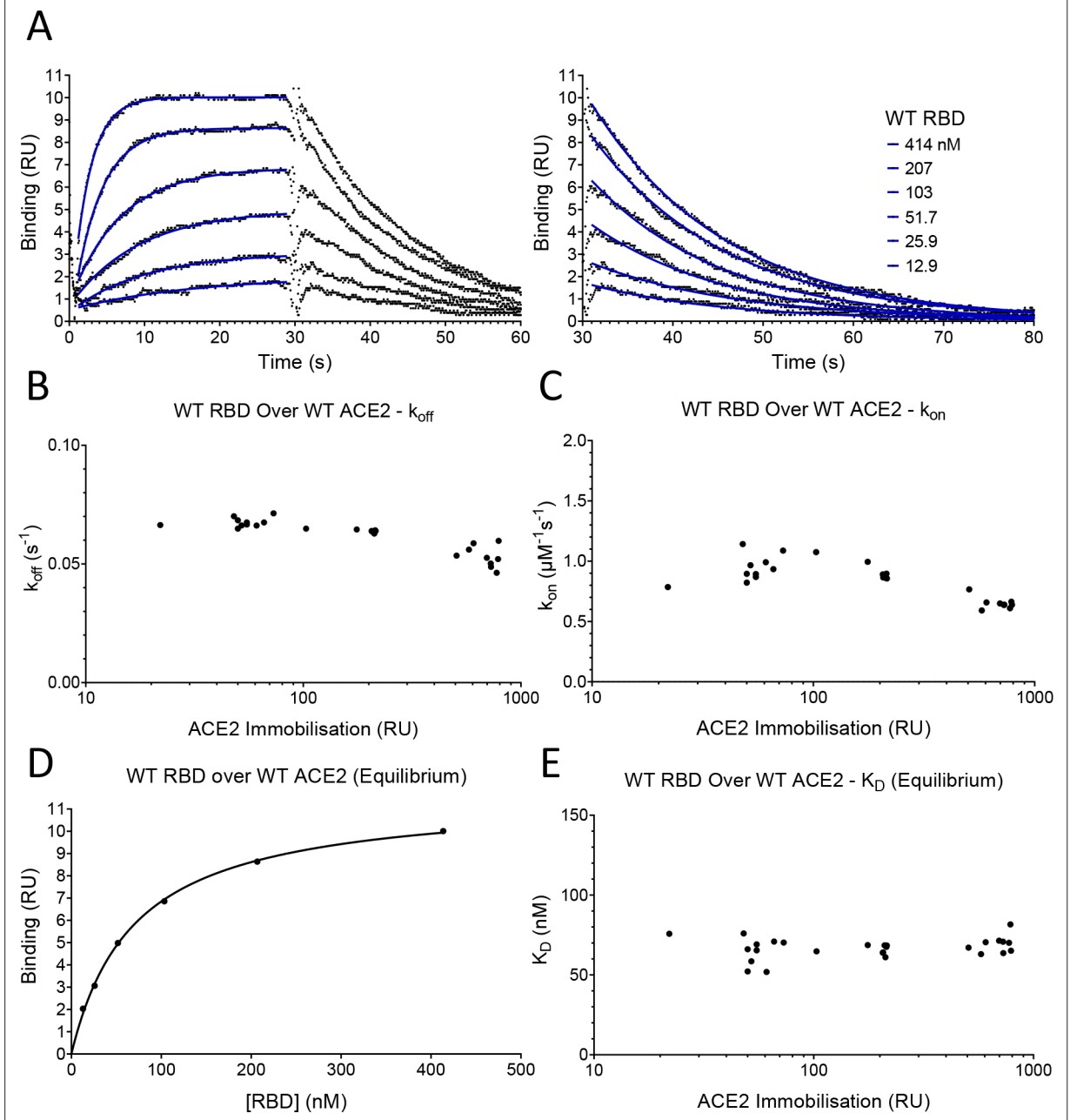

**Figure 2.** SPR analysis. (**A**) Overlay of traces showing association and dissociation when WT RBD is injected for 30 s at the indicated concentration over immobilised WT ACE2. The right panel shows an expanded view of the dissociation phase. The blue lines show the fits used for determining the $k_{on}$ and $k_{off}$. The $k_{on}$ was determined as described in *Figure 2—figure supplement 2*. The $k_{off}$ (**B**) and $k_{on}$ (**C**) values measured at different levels of immobilised ACE2 are shown. (**D**) The equilibrium $K_D$ was determined by plotting the binding at equilibrium against [RBD] injected. Data from experiment shown in (**A**). (**E**) The equilibrium $K_D$ measured at different levels of immobilised ACE2 are shown.

The online version of this article includes the following source data and figure supplement(s) for figure 2:

**Source data 1.** Source data for *Figure 2*.

**Figure supplement 1.** Protein purification.

**Figure supplement 2.** Determining the $k_{on}$ and $k_{off}$.

**Figure supplement 2—source data 1.** Source data for *Figure 2—figure supplement 2*.

**Table 1.** Affinity and kinetic data for RBD variants and ACE2 variants.

Mean and SD of the $k_{off}$, $k_{on}$, calculated $K_D$, and equilibrium $K_D$ values for all RBD variants binding all ACE2 variants. For most measurements n = 3, the exceptions were RBD WT/ACE2 WT equilibrium $K_D$ measurements (n = 24) and other RBD WT measurements (n = 6). UK2 refers to the VOC-202102–02 variant.

| | $k_{off}$ (s$^{-1}$) | SD | $k_{on}$ (μM$^{-1}$ s$^{-1}$) | SD | $K_D$ calc. (nM) | SD | $K_D$ equi. (nM) | SD |
|---|---|---|---|---|---|---|---|---|
| **RBD over WT ACE2** | | | | | | | | |
| WT | 0.0668 | 0.00113 | 0.90 | 0.05 | 74.4 | 4.0 | 62.6 | 7.7 |
| K417N | 0.177 | 0.00416 | 0.49 | 0.05 | 364 | 29 | 349 | 10 |
| K417T | 0.126 | 0.00510 | 0.55 | 0.04 | 230 | 23 | 226 | 19 |
| S477N | 0.0348 | 0.00037 | 0.81 | 0.03 | 42.9 | 2.1 | 42.6 | 3.0 |
| E484K | 0.0818 | 0.00183 | 1.54 | 0.03 | 53.1 | 1.7 | 52.6 | 2.0 |
| N501Y (Alpha) | 0.0111 | 0.00017 | 1.59 | 0.04 | 7.0 | 0.25 | 5.5 | 2.4 |
| K417N/E484K | 0.251 | 0.00799 | 1.02 | 0.07 | 247 | 23 | 251 | 23 |
| K417T/E484K | 0.168 | 0.00573 | 1.10 | 0.05 | 153 | 12 | 147 | 8.6 |
| E484K/N501Y (UK2) | 0.0118 | 0.00037 | 2.33 | 0.10 | 5.1 | 0.36 | 3.7 | 2.7 |
| K417N/E484K/ N501Y (Beta) | 0.0291 | 0.00076 | 1.46 | 0.06 | 20.0 | 0.70 | 17.4 | 3.1 |
| K417T/ E484K/N501Y (Gamma) | 0.0211 | 0.00021 | 1.56 | 0.07 | 13.5 | 0.45 | 12.2 | 3.4 |
| **RBD over S19P ACE2** | | | | | | | | |
| WT | 0.0298 | 0.00039 | 1.50 | 0.12 | 20.0 | 1.3 | 30.5 | 2.2 |
| K417N | 0.0782 | 0.00284 | 0.72 | 0.04 | 108 | 2.8 | 129 | 8.2 |
| K417T | 0.0521 | 0.00196 | 0.69 | 0.02 | 75.8 | 4.7 | 87.8 | 7.0 |
| S477N | 0.0257 | 0.00016 | 1.05 | 0.07 | 24.6 | 1.7 | 30.3 | 2.7 |
| E484K | 0.0325 | 0.00031 | 2.02 | 0.08 | 16.2 | 0.55 | 20.8 | 1.3 |
| N501Y (Alpha) | 0.0051 | 0.00004 | 2.31 | 0.09 | 2.2 | 0.09 | 3.5 | 0.4 |
| K417N/E484K | 0.0961 | 0.00198 | 1.28 | 0.11 | 75.6 | 7.1 | 91.3 | 6.5 |
| K417T/E484K | 0.0660 | 0.00255 | 1.45 | 0.03 | 45.5 | 2.5 | 53.8 | 1.5 |
| E484K/N501Y (UK2) | 0.0051 | 0.00008 | 3.10 | 0.10 | 1.7 | 0.05 | 3.4 | 0.4 |
| K417N/E484K/ N501Y (Beta) | 0.0122 | 0.00009 | 2.16 | 0.03 | 5.7 | 0.07 | 10.4 | 1.2 |
| K417T/ E484K/N501Y (Gamma) | 0.0085 | 0.00007 | 2.11 | 0.05 | 4.0 | 0.07 | 6.1 | 1.3 |
| **RBD over K26R ACE2** | | | | | | | | |
| S477N | 0.0240 | 0.00009 | 1.07 | 0.05 | 22.6 | 1.1 | 33.4 | 1.3 |
| WT | 0.0500 | 0.00062 | 1.60 | 0.16 | 31.4 | 2.6 | 48.8 | 2.5 |
| K417N | 0.154 | 0.00789 | 0.88 | 0.07 | 175 | 8.1 | 237 | 15 |
| K417T | 0.101 | 0.00079 | 0.81 | 0.12 | 127 | 17.4 | 154 | 2.8 |

*Table 1 continued on next page*

*Table 1 continued*

| | $k_{off}$ (s⁻¹) | SD | $k_{on}$ (µM⁻¹ s⁻¹) | SD | $K_D$ calc. (nM) | SD | $K_D$ equi. (nM) | SD |
|---|---|---|---|---|---|---|---|---|
| S477N | 0.0240 | 0.00009 | 1.07 | 0.05 | 22.6 | 1.1 | 33.4 | 1.3 |
| E484K | 0.0587 | 0.00109 | 2.03 | 0.03 | 28.9 | 1.0 | 35.9 | 1.5 |
| N501Y (Alpha) | 0.0081 | 0.00002 | 2.34 | 0.09 | 3.5 | 0.15 | 7.5 | 1.5 |
| K417N/E484K | 0.191 | 0.00481 | 1.48 | 0.15 | 130 | 9.4 | 166 | 11 |
| K417T/E484K | 0.135 | 0.00407 | 1.53 | 0.02 | 88.0 | 3.9 | 105 | 0.7 |
| E484K/N501Y (UK2) | 0.0085 | 0.00018 | 3.06 | 0.23 | 2.8 | 0.17 | 6.4 | 0.3 |
| K417N/E484K/ N501Y (Beta) | 0.0234 | 0.00040 | 2.13 | 0.05 | 11.0 | 0.28 | 18.7 | 2.0 |
| K417T/ E484K/N501Y (Gamma) | 0.0164 | 0.00028 | 2.21 | 0.06 | 7.4 | 0.33 | 15.3 | 0.8 |

was equal to the sum of the ΔΔG measured for ACE2 variant/RBD WT and ACE WT/RBD variant interactions. This is depicted as the difference between the measured and predicted ΔΔG for inter-actions between ACE2 and RBD variants (ΔΔΔG in *Figure 4B,C*, *Dejnirattisai et al., 2021*). In most cases, ΔΔΔG values were close to zero, indicating that the effects of these mutations were largely independent. The one exception was the combination of ACE2 S19P and RBD S477N variants, where the measured value was significantly lower than the predicted value (*Figure 4B*), indicating that these mutations were not independent. This is consistent with the fact that the ACE2 residue S19 is adjacent to RBD residue S477 in the contact interface (*Figure 1C*). An important consequence of this is that the S477N mutation increased the affinity of RBD for ACE2 WT but decreased its affinity for ACE2 S19P.

## Discussion

While our finding that the SARS-CoV-2 RBD binds ACE2 with an affinity of $K_D$ 74 nM at 37 °C is consistent with previous studies ($K_D$ 6–133 nM) (*Laffeber et al., 2021*; *Lei et al., 2020*; *Liu et al., 2021*; *Shang et al., 2020*; *Supasa et al., 2021*; *Wrapp et al., 2020*; *Zhang et al., 2021*; *Zhang et al., 2020*), the rate constants that we measured ($k_{on}$ 0.9 µM⁻¹.s⁻¹ and $k_{off}$ 0.067 s⁻¹) were faster than all previous reports. One likely reason for this is that previous measurements were performed at a lower temperature, which almost always decreases rate constants. While some studies stated that binding constants were measured at 25 °C (*Laffeber et al., 2021*; *Zhang et al., 2020*), most studies did not report the temperature, suggesting that they were performed at room temperature or the standard instrument temperature (20°C–25°C). A second likely reason is that previous kinetic studies were performed under conditions in which the rate of diffusion of soluble molecule to the sensor surface limits the association rate, and rebinding of dissociated molecules to the surface reduces the measured dissociation rate. These are known pitfalls of both techniques used in these studies, surface plasmon resonance (*Myszka, 1997*), and bilayer interferometry (*Abdiche et al., 2008*). In the present study, we avoided these issues by immobilising a very low level of ligand on the sensor surface. A third possible reason is that the proteins were aggregated, which can cause problems even when aggre-gates are a very minor contaminant (*van der Merwe and Barclay, 1996*). The presence of aggregates results in complex binding kinetics, which can be excluded if the simple 1:1 Langmuir binding model fits the kinetic data. While this was demonstrated in the present study, and some previous studies (*Shang et al., 2020*; *Wrapp et al., 2020*; *Zhang et al., 2021*), such fits were not shown in all studies, one of which reported more than 20-fold slower kinetics than reported here (*Lei et al., 2020*; *Supasa et al., 2021*).

The RBD mutations that we selected for analysis have all emerged independently and became dominant in a region at least once in different lineages, suggesting that they provide a selective advantage. Our finding that the N501Y, E484K, and S477N all increase the binding affinity of RBD for ACE2 raises the question as to whether this contributed to their selection. Several lines of evidence

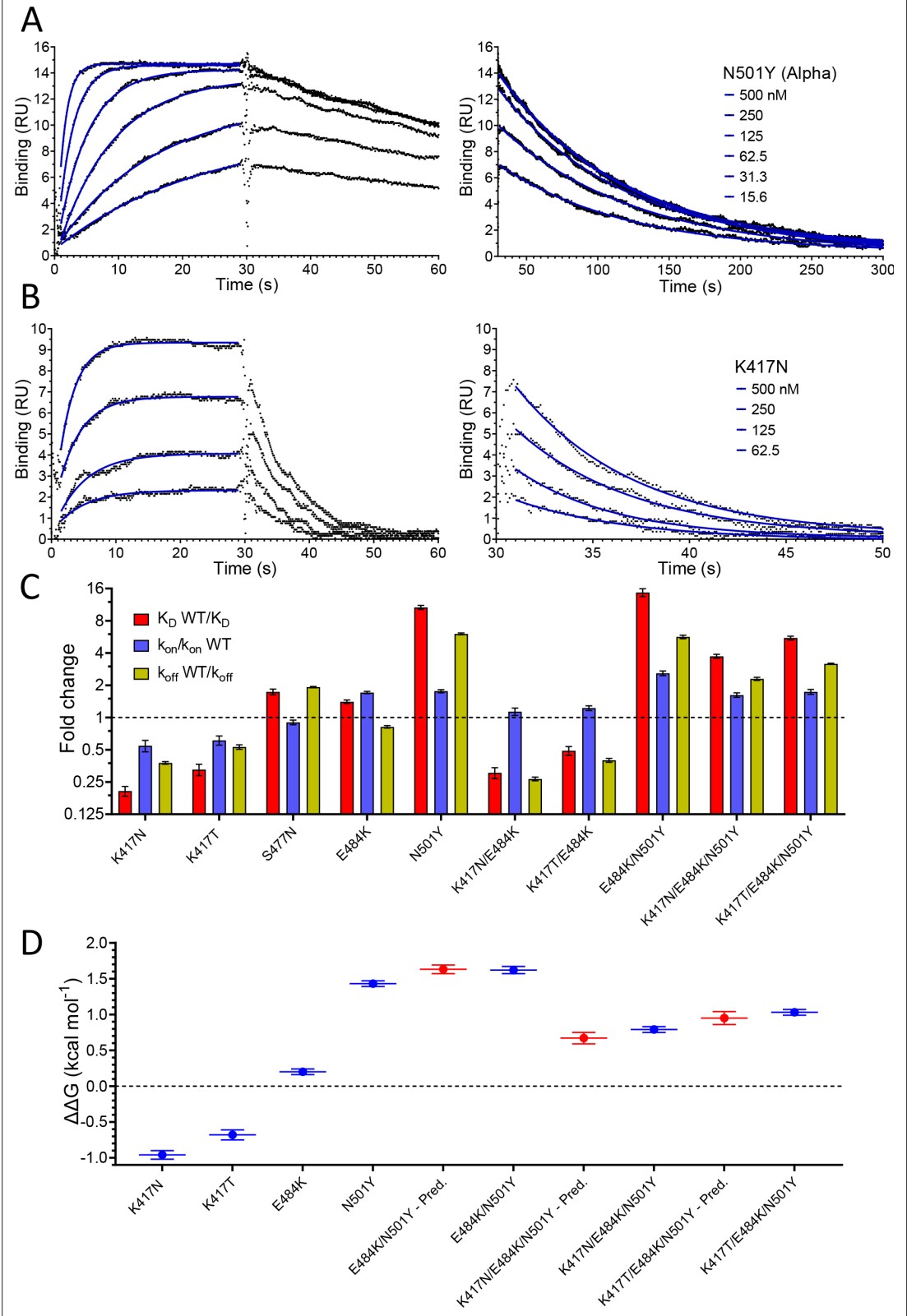

**Figure 3.** Effect of RBD mutations on binding to WT ACE2. Overlay of traces showing association and dissociation of N501Y (**A**) and K417N (**B**) RBD variants when injected at a range of concentrations over immobilised WT ACE2. The right panels show an expanded view of the dissociation phase. The blue lines show fits used for determining the $k_{on}$ and $k_{off}$. (**C**) The fold change relative to WT RBD of the calculated $K_D$, $k_{on}$, and $k_{off}$ for binding of the indicated RBD variants to immobilised WT ACE2 (error bars show SD, n = 3). Representative sensorgrams from all mutants shown in *Figure 3—figure*

*Figure 3 continued on next page*

*Figure 3 continued*

*supplement 2*, and the mean values from multiple repeats are in *Table 1*. (**D**) The blue lines show the measured ΔΔG for indicated RBD variants. The red lines show the predicted ΔΔG for the RBD variants with multiple mutations, which were calculated by adding ΔΔG values for single mutation variants (error bars show SD, n = 3).

The online version of this article includes the following source data and figure supplement(s) for figure 3:

**Source data 1.** Source data for *Figure 3*.

**Figure supplement 1.** Mass transport controls for RBD.

**Figure supplement 1—source data 1.** Source data for *Figure 3—figure supplement 1*.

**Figure supplement 2.** Representative SPR data for RBD variants binding to WT ACE2.

**Figure supplement 2—source data 1.** Source data for *Figure 3—figure supplement 2*.

suggest that enhancing the Spike/ACE2 interaction would be advantageous for the virus. First, the virus has spread only very recently to humans from another mammalian host, providing insufficient time for optimisation of the affinity. Second, epidemiological studies have suggested that the Alpha variant, which has the N501Y mutation, has enhanced transmissibility (*Volz et al., 2021a*; *Washington et al., 2021*). Finally, a SARS-CoV-2 variant with the Spike mutation D614G, which increases its activity by stabilising it following furin cleavage (*Zhang et al., 2021*; *Zhang et al., 2020*), rapidly became dominant globally after it emerged (*Korber et al., 2020*; *Volz et al., 2021b*). Taken together, these findings suggest that the WT Spike/ACE2 interaction is limiting for transmission and that mutations that enhance it, including the N501Y, E484K, and S477N mutations, could provide a selective advantage by increasing transmissibility. This raises two questions. First, will other RBD mutations appear in SARS-CoV-2 which further enhance transmission? This seems likely, given that a large number of RBD mutations have been identified that increase the RBD/ACE2 affinity (*Starr et al., 2020*; *Zahradník et al., 2021*). Second, will combinations of existing mutations be selected because they further increase the affinity? While the appearance E484K, together with N501Y in three lineages (Alpha, Beta, and Gamma), supports this, it is also possible that E484K was selected because it disrupts antibody neutraliaation, as discussed below.

Our affinity and kinetic data on RBD variants are broadly consistent with some (*Laffeber et al., 2021*; *Liu et al., 2021*), but not all (*Dejnirattisai et al., 2021*; *Zhou et al., 2021a*), recent reports on the K417T/N, N501Y, and E484K variants. One caveat to our study is that we used monomeric forms of RBD and ACE2. The native Spike protein is a trimer and has several other domains, including the nearby N-teminal domain, and the native ACE2 protein can exist as a dimer (*Yan et al., 2020*).

**Table 2.** ΔΔG for RBD variants binding to ACE2 variants.

Mean and SD of ΔΔG (n = 3, kcal/mol) were determined as described in Materials and methods using the calculated $K_D$ values in *Table 1*. UK2 refers to the VOC-202102–02 variant.

| RBD variant | Ace2 wt | | Ace2 s19p | | Ace2 k26r | |
|---|---|---|---|---|---|---|
| | ΔΔG | SD | ΔΔG | SD | ΔΔG | SD |
| WT | 0.00 | 0.00 | 0.79 | 0.05 | 0.52 | 0.06 |
| K417N | –0.96 | 0.06 | –0.23 | 0.04 | –0.52 | 0.04 |
| K417T | –0.68 | 0.07 | –0.01 | 0.05 | –0.32 | 0.09 |
| S477N | 0.33 | 0.04 | 0.67 | 0.05 | 0.72 | 0.04 |
| E484K | 0.20 | 0.04 | 0.92 | 0.04 | 0.57 | 0.04 |
| N501Y (Alpha) | 1.43 | 0.04 | 2.13 | 0.04 | 1.86 | 0.04 |
| K417N/E484K | –0.72 | 0.07 | –0.01 | 0.07 | –0.34 | 0.06 |
| K417T/E484K | –0.43 | 0.06 | 0.30 | 0.05 | –0.10 | 0.04 |
| E484K/N501Y (UK2) | 1.62 | 0.05 | 2.30 | 0.04 | 1.98 | 0.05 |
| K417N/E484K/N501Y (Beta) | 0.79 | 0.04 | 1.56 | 0.03 | 1.16 | 0.04 |
| K417T/E484K/N501Y (Gamma) | 1.03 | 0.04 | 1.76 | 0.03 | 1.39 | 0.04 |

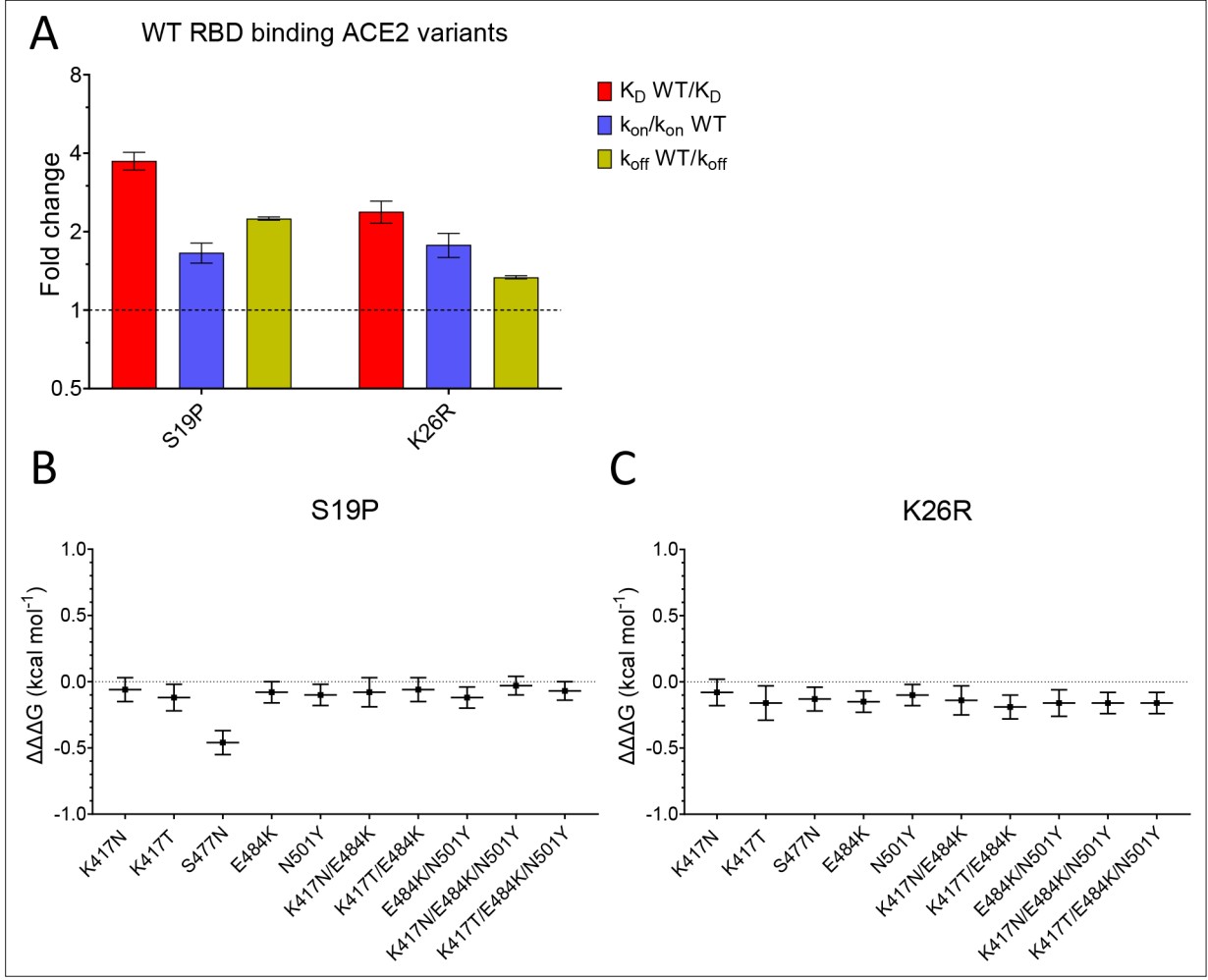

**Figure 4.** Effect of mutations in ACE2. (**A**) The fold change relative to WT ACE2 of the calculated $K_D$, $k_{on}$, and $k_{off}$ for the interaction of WT RBD and the indicated ACE2 variants (error bars show SD, n = 3). (**B, C**) Show the difference ($\Delta\Delta\Delta G$) between the measured and predicted $\Delta\Delta G$ for S19P (**B**) and K26R (**C**) ACE2 variants binding to the indicated RBD variants, calculated from data in *Table 2*. The predicted $\Delta\Delta G$ values for each variant RBD/variant ACE2 interaction were calculated from the sum of the $\Delta\Delta G$ for the ACE2 variant binding WT RBD and the $\Delta\Delta G$ for the RBD variant binding WT ACE2 (*Table 2*).

The online version of this article includes the following source data and figure supplement(s) for figure 4:

**Source data 1.** Source data for *Figure 4*.

**Figure supplement 1.** Representative SPR data for WT RBD binding ACE2 variants.

**Figure supplement 1—source data 1.** Source data for *Figure 4—figure supplement 1*.

Because of these differences, our analysis may not detect the full effects of RBD and ACE2 mutations on the Spike/ACE2 interaction. A second caveat is that we have not examined the effect of these mutations on viral attachment to cells.

Studies of other enveloped viruses, including SARS-CoV, suggest that increases in affinity of viral ligands for their cellular receptors can increase cell infection and disease severity (*Hasegawa et al., 2007*; *Li et al., 2005*). One study found that increasing this affinity enabled the virus to infect cells with lower receptor surface density (*Hasegawa et al., 2007*). It follows that increases in affinity could increase the number of host tissues infected, which could increase the severity of disease (*Cao and Li, 2020*) and/or increase the viral load in the upper respiratory tract (*Hoffmann et al., 2020*; *Wölfel et al., 2020*), thereby increasing spread.

Another mechanism by which mutations of RBD could provide a selective advantage is through evasion of immune responses. This is supported by the observation that neutralising antibodies present in those infected by or vaccinated against SARS-CoV-2 primarily target the RBD (*Garcia-Beltran et al., 2021*; *Greaney et al., 2021a*; *Rogers et al., 2020*). Furthermore, two variants with

RBD mutations that abrogate antibody neutralisation, Beta and Gamma, became dominant in regions with very high levels of prior SARS-CoV-2 infection (*Cele et al., 2021*; *Dejnirattisai et al., 2021*; *Hoffmann et al., 2021*; *Sabino et al., 2021*; *Tegally et al., 2021*; *Zhou et al., 2021b*). Both lineages include the N501Y mutation, but this appears to have modest effects on antibody neutralisation (*Greaney et al., 2021a*; *Greaney et al., 2021b*). In contrast, the E484K mutation, also present in both variants, potently disrupts antibody neutralisation (*Greaney et al., 2021a*; *Greaney et al., 2021b*). Our finding that the K417N/T mutations present in Beta and Gamma variants decrease the affinity of RBD for ACE2 suggests that they were selected because they facilitate immune escape. Indeed, mutations of K417 can block antibody neutralisation, albeit less effectively than E484K (*Greaney et al., 2021a*; *Greaney et al., 2021b*; *Wang et al., 2021*). It is notable that these affinity-reducing K417N/T mutations have only emerged together with mutations (N501Y and E484K) that increase the affinity of RBD for ACE2, suggesting a cooperative effect between mutations that enhance immune escape and mutations that increase affinity.

The effect of the increased affinity for SARS-CoV-2 Spike RBD of the K26R and S19P ACE2 mutants is less clear. The evidence summarised above that WT RBD/ACE2 binding is limiting for SARS-CoV-2 transmission, suggest that carriers of these ACE2 variants will be at greater risk of infection and/or severe disease. However, in contrast to SARS-CoV-2 RBD mutations, the effects of ACE2 variants are primarily relevant to the carriers of these mutations. A preliminary analysis (*MacGowan et al., 2021*) suggests that the carriers of the K26R ACE allele might be at increased risk of severe disease, but the findings did not reach statistical significance, and further studies are required.

The interaction that we identified between the RBD S477N and ACE2 S19P mutants highlights the importance of considering variation in the host population when studying the evolution of viral variants. In this case, the opposite effect of the RBD S477N mutation on its affinity for ACE2 S19P (decreased), compared with ACE2 WT (increased), suggests that this RBD variant may have a selective disadvantage amongst carriers of the ACE2 S19P variant, in contrast to those with ACE2 WT, where it appears to be advantageous. However, the low frequency of this variant means that this is unlikely to be important at a population level and will be difficult to detect.

It is noteworthy that the two most common ACE2 variants are in positions on ACE2 with no known functional activity. This raises the question as to whether these mutations are a remnant of historic adaption to pathogens that utilised this portion of ACE2. The fact that ACE2 S19P mutation is largely confined to African/African-American populations, suggests that it is more recent than K26R and/or selected by pathogen(s) confined to the African continent.

# Materials and methods

**Key resources table**

| Reagent type (species) or resource | Designation | Source or reference | Identifiers | Additional information |
|---|---|---|---|---|
| Transfected construct (human) | ACE2 WT | Oxford Protein Production Facility-UK | pOPINTTGneo_ACE2-BAP | T |
| Transfected construct (human) | ACE2 S19P; ACE2 K26R | This paper | | Available from authors |
| Transfected construct (*SARS-CoV-2*) | RBD WT | BEI Resources, NIH | NR-52309 | pCAGG plasmid |
| Transfected construct (*SARS-CoV-2*) | RBD K417N; RBD RBD K417T; RBD S477N; RBD E484K; RBD N501Y; RBD K417N/E484K; RBD K417T/E484K; RBD beta; RBD gamma | This paper | | pCAGG plasmid. Available from authors |
| Transfected construct (human) | pTT3-BirA-FLAG | Addgene | RRID:Addgene_64395 | Cotranfected for in-cell biotinylation |
| Peptide, recombinant protein | ACE2 WT; ACE2 S19P; ACE2 K26R | This paper | | Expressed in HEK293 cells and purified |

*Continued on next page*

*Continued*

| Reagent type (species) or resource | Designation | Source or reference | Identifiers | Additional information |
|---|---|---|---|---|
| Peptide, recombinant protein | RBD WT; RBD K417N; RBD K417T; RBD S477N; RBD E484K; RBD N501Y; RBD K417N/E484K; RBD K417T/E484K; RBD beta; RBD gamma | This paper | | Expressed in HEK293 cells and purified |
| Antibody | anti-human ACE2 (mouse monoclonal) | NOVUS Biologicals | AC384 | (5 µg/mL) |
| Cell line (human) | FreeStyle HEK293F Cells | ThermoFisher Scientific | RRID:CVCL_D603 | |
| Chemical compound, drug | FreeStyle MAX Reagent | ThermoFisher | 16447100 | |
| Chemical compound, drug | FreeStyle 293 Expression Medium | ThermoFisher | 12338018 | |
| commercial assay or kit | QuikChange II XL | Agilent | 200,521 | |
| Commercial assay or kit | Amine coupling kit | Cytiva | BR100050 | |
| Software, algorithm | GraphPad | Prism | Version 9 | |
| Other | CM5 sensor chips | Cytiva | 29149603 | |

## ACE2 and RBD variant constructs

The plasmid used to express soluble ACE2 WT (pOPINTTGneo_ACE2-BAP), which was kindly provided by Ray Owens (Oxford Protein Production Facility-UK), encoded the following protein:

> STIEEQAKTFLDKFNHEAEDLFYQSSLASWNYNTNITEENVQNMNNAGDKWSAFLKEQSTLAQM
> YPLQEIQNLTVKLQLQALQQNGSSVLSEDKSKRLNTILNTMSTIYSTGKVCNPDNPQECLLLEP
> GLNEIMANSLDYNERLWAWESWRSEVGKQLRPLYEEYVVLKNEMARANHYEDYGDYWRGD
> YEVNGVDGYDYSRGQLIEDVEHTFEEIKPLYEHLHAYVRAKLMNAYPSYISPIGCLPAHLLGDM
> WGRFWTNLYSLTVPFGQKPNIDVTDAMVDQAWDAQRIFKEAEKFFVSVGLPNMTQGFWEN
> SMLTDPGNVQKAVCHPTAWDLGKGDFRILMCTKVTMDDFLTAHHEMGHIQYDMAYAAQPF
> LLRNGANEGFHEAVGEIMSLSAATPKHLKSIGLLSPDFQEDNETEINFLLKQALTIVGTLPFTYMLEK
> WRWMVFKGEIPKDQWMKKWWEMKREIVGVVEPVPHDETYCDPASLFHVSNDYSFIRYYTRTLYQ
> FQFQEALCQAAKHEGPLHKCDISNSTEAGQKLFNMLRLGKSEPWTLALENVVGAKNMNVRPLLN
> YFEPLFTWLKDQNKNSFVGWSTDWSPYADLNDIFEAQKIEWHEKHHHHHH

The carboxy-terminal end has a biotin acceptor peptide (underlined) followed by an oligohistidine tag.

The pCAGG plasmid used to express the RBD WT construct (*Amanat et al., 2020*) encoded the following protein:

> RVQPTESIVRFPNITNLCPFGEVFNATRFASVYAWNRKRISNCVADYSVLYNSASFSTFKCYGVSPTK
> LNDLCFTNVYADSFVIRGDEVRQIAPGQTGKIADYNYKLPDDFTGCVIAWNSNNLDSKVGGNYN
> YLYRLFRKSNLKPFERDISTEIYQAGSTPCNGVEGFNCYFPLQSYGFQPTNGVGYQPYRVVVLSFELL
> HAPATVCGPKKSTNLVKNKCVNFHHHHHH

The carboxy-terminal end has an oligohistidine tag.

ACE2 and RBD point mutations were introduced into these plasmid constructs using the Agilent QuikChange II XL Site-Directed Mutagenesis Kit following the manufacturer's instructions. The primers were designed using the Agilent QuikChange primer design web program.

## HEK293F cell transfection

Cells were grown in FreeStyle 293 Expression Medium (ThermoFisher Scientific, 12338018) in a 37 °C incubator with 8 % $CO_2$ on a shaking platform at 130 rpm. Cells were passaged every 2–3 days with the suspension volume always kept below 33.3 % of the total flask capacity. The cell density was kept between 0.5 and 2 million per ml. Before transfection cells were counted to check that cell viability

was above 95 %, and the density was adjusted to 1.0 million per ml. For 100 ml transfection, 100 µl FreeStyle MAX Reagent (ThermoFisher Scientific, 16447100) was mixed with 2 ml Opti-MEM (ThermoFisher Scientific, 51985034) for 5 min. During this incubation, 100 µg of expression plasmid was mixed with 2 ml Opti-MEM (or in situ biotinylation of ACE2 90 µg of expression plasmid was mixed with 10 µg of expression plasmid encoding the BirA enzyme). The DNA was then mixed with the MAX Reagent and incubated for 25 min before being added to the cell culture. For ACE2 in situ biotinylation, biotin was added to the cell culture at a final concentration of 50 µM. The culture was left for 5 days for protein expression to take place.

## Protein purification

Cells were harvested by centrifugation and the supernatant collected and filtered through a 0.22 µm filter. Imidazole was added to a final concentration of 10 mM and PMSF added to a final concentration of 1 mM; 1 ml of Ni-NTA Agarose (Qiagen; 30310) was added per 100 ml of supernatant and the mix was left on a rolling platform at 4 °C overnight. The mix was poured through a gravity flow column to collect the Ni-NTA Agarose. The Ni-NTA Agarose was washed three times with 25 ml of wash buffer (50 mM $NaH_2PO_4$, 300 mM NaCl, and 20 mM imidazole at pH 8). The protein was eluted with elution buffer (50 mM $NaH_2PO_4$, 300 mM NaCl, and 250 mM imidazole at pH 8). The protein was concentrated, and buffer exchanged into size exclusion buffer (25 mM $NaH_2PO_4$ and 150 mM NaCl at pH 7.5) using a protein concentrator with a 10,000 molecular weight cut-off. The protein was concentrated down to less than 500 µl and loading onto a Superdex 200 10/300 GL (Cytiva, 17-5175-01) size exclusion column (*Figure 2—figure supplement 1*). Fractions corresponding to the desired peak were pooled and frozen at –80 °C. Samples from all observed peaks were analysed on a reducing SDS–PAGE gel (*Figure 2—figure supplement 1*).

## Surface plasmon resonance

RBD binding to ACE2 was analysed on a Biacore T200 instrument (Cytiva) at 37 °C and a flow rate of 30 µl/min. Running buffer was HBS-EP (Cytiva, BR100669). Streptavidin was coupled with a CM5 sensor chip (Cytiva, 29149603) using an amine coupling kit (Cytiva, BR100050) to near saturation, typically 10,000–12,000 response units (RU). Biotinylated ACE2 WT and variants were injected into the experimental flow cells (FC2–FC4) for different lengths of time to produce desired immobilisation levels (20–800 RU). FC1 was used as a reference and contained streptavidin only. Excess streptavidin was blocked with two 40 s injections of 250 µM biotin (Avidity). Before RBD injections, the chip surface was conditioned with eight injections of the running buffer. A dilution series of RBD was then injected in all FCs. Buffer alone was injected after every two or three RBD injections. The length of all injections was 30 s, and dissociation was monitored for 180–670 s. The background response measured in FC1 was subtracted from the response in the other three FCs. In addition, the responses measured during buffer injections closest in time were subtracted. Such double-referencing improves data quality when binding responses are low as needed to obtain accurate kinetic data (*Myszka, 1999*). At the end of each experiment, an ACE2-specific mouse monoclonal antibody (NOVUS Biologicals, AC384) was injected at 5 µg/ml for 10 min to confirm the presence and relative amounts of immobilised ACE2.

## Data analysis

Double-referenced binding data was fitted using GraphPad Prism. The $k_{off}$ was determined by fitting a mono-exponential decay curve to data from the dissociation phase of each injection. The $k_{off}$ from four to six RBD injections was averaged (*Figure 2—figure supplement 2A*). The $k_{on}$ was determined by first fitting a mono-exponential association curve to data from the association phase, yielding the $k_{obs}$, and then plotting the $k_{obs}$ vs the concentration of RBD and performing a linear fit of the equation $k_{obs}$ = $k_{on}*[RBD]+ k_{off}$ to this data (*Figure 2—figure supplement 2B*), using the $k_{off}$ determined as above to constrain the fit.

The $K_D$ was either calculated (calculated $K_D = k_{off}/k_{on}$) or measured directly (equilibrium $K_D$) as follows. Equilibrium binding levels at a given [RBD] were determined from the fit of the mono-exponential association phase model to the association phase data. These equilibrium binding levels were plotted against [RBD], and a fit of the simple 1:1 Langmuir binding model to this data was used to determine the equilibrium $K_D$ (*Figure 2D*).

$\Delta G$ for each affinity measurement was calculated from the relationship $\Delta G = R*T*\ln K_D$, where $R$ = 1.987 cal mol$^{-1}$ K$^{-1}$, T = 310.18 K, and $K_D$ is in units M. $\Delta \Delta G$ values (*Table 2* and *Figure 3D*) were calculated for each mutant from the relationship $\Delta \Delta G = \Delta G_{WT} - DG_M$. The predicted $\Delta \Delta G$ for interactions with multiple mutants were calculated by adding the single mutant $\Delta \Delta G$ values (*Figure 3D*). The difference between the measured and predicted $\Delta \Delta G$ ($\Delta \Delta \Delta G$) for interactions between the ACE2 and RBD mutants was calculates as $\Delta \Delta \Delta G$ = measured $\Delta \Delta G$ – predicted $\Delta \Delta G$ (*Figure 4B*).

All errors represent standard deviations and errors for calculated values were determined by error propagation.

## Acknowledgements

We thank Johannes Pettmann for help with protein expression and Anna Huhn for help with data analysis. OD is supported by a Wellcome Trust Senior Fellowship in Basic Biomedical Sciences (207537/Z/17/Z828). SM and GB are supported by Biotechnology and Biological Sciences Research Council Grants (BB/J019364/1 and BB/R014752/1) and a Wellcome Trust Biomedical Resources Grant (101651/Z/13/Z).

## Additional information

### Competing interests

P Anton van der Merwe: Own shares in BioNTech SE. The other authors declare that no competing interests exist.

### Funding

| Funder | Grant reference number | Author |
| --- | --- | --- |
| Wellcome Trust | 207537/Z/17/Z828 | Mikhail A Kutuzov<br>Omer Dushek |
| Biotechnology and Biological Sciences Research Council | BB/J019364/1 | Stuart A MacGowan<br>Geoffrey John Barton |
| Biotechnology and Biological Sciences Research Council | BB/R014752/1 | Stuart A MacGowan<br>Geoffrey John Barton |
| Wellcome Trust | 101651/Z/13/Z). | Stuart A MacGowan<br>Geoffrey John Barton |

The funders had no role in study design, data collection and interpretation, or the decision to submit the work for publication.

### Author contributions

Michael I Barton, Data curation, Formal analysis, Investigation, Methodology, Project administration, Visualization, Writing – original draft, Writing – review and editing; Stuart A MacGowan, Conceptualization, Investigation, Visualization, Writing – review and editing; Mikhail A Kutuzov, Investigation, Methodology, Supervision; Omer Dushek, Conceptualization, Funding acquisition, Resources, Writing – review and editing; Geoffrey John Barton, Conceptualization, Resources, Supervision, Writing – review and editing; P Anton van der Merwe, Conceptualization, Formal analysis, Funding acquisition, Supervision, Writing – original draft, Writing – review and editing

### Author ORCIDs

Michael I Barton ⓘ http://orcid.org/0000-0002-9263-6481
Stuart A MacGowan ⓘ http://orcid.org/0000-0003-4233-5071
Mikhail A Kutuzov ⓘ http://orcid.org/0000-0003-3386-4350
Omer Dushek ⓘ http://orcid.org/0000-0001-5847-5226
Geoffrey John Barton ⓘ http://orcid.org/0000-0002-9014-5355
P Anton van der Merwe ⓘ http://orcid.org/0000-0001-9902-6590

**Decision letter and Author response**
Decision letter https://doi.org/10.7554/eLife.70658.sa1
Author response https://doi.org/10.7554/eLife.70658.sa2

## Additional files

### Supplementary files
• Transparent reporting form

### Data availability
All data generated and analysed during this study are included in the manuscript and supporting files.

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
