## [Decision Letter]

**Acceptance summary:**

This manuscript is of interest to infectious disease specialists, biochemists and biophysicists working on SARS-CoV-2. It provides detailed and rigorous affinity and kinetics analyses of the effect of several biologically relevant amino acid substitutions in the SARS-CoV-2 receptor binding domain and ACE2 as measured by surface plasmon resonance with recombinant proteins. Although the study is limited to a binding analysis of monomeric protein domains, the data may contribute to understand the rapid emergence of SARS-CoV-2 variants with substitutions in the spike protein.

**Decision letter after peer review:**

Thank you for submitting your article "Effects of common mutations in the SARS-CoV-2 Spike RBD domain and its ligand the human ACE2 receptor on binding affinity and kinetics" for consideration by *eLife*. Your article has been reviewed by 3 peer reviewers, including Ron AM Fouchier as Reviewing Editor and Reviewer #1, and the evaluation has been overseen by Jos van der Meer as the Senior Editor.

Essential revisions:

No new experiments are required but numerous suggestions have been made by the 3 reviewers to correct or clarify the text.

*Reviewer #1 (Recommendations for the authors):*

Several countries have requested to not use country names in virus lineage descriptions, to avoid stigmatization. I would suggest to (maximally) once in the manuscript mention for each lineage where it was first detected, and afterwards refrain from using country names and country codes. In many tables and figures and legends and texts, these names recur (e.g. table 2, Figure S2 and elsewhere).

Line 20. SARS-CoV-2 is not the second coronavirus that can induce a severe acute respiratory syndrome; all human coronaviruses can do that, and so can SARS-CoV and MERS-CoV.

Line 22. This is a matter of debate. I guess the authors only take infectious disease pandemics into account and do not consider the HIV pandemic to have unfolded in the past 100 years?

Line 61 and elsewhere. The authors mention "physiological temperature". It would be useful throughout the text to add the exact temperature used (37C) in statements like this. The temperature in the human airways is not the same everywhere (less than 37C in the upper airways). It is better to be exact.

On several occasions in the manuscript, the authors use "transmissivity" where they probably mean "transmissibility".

I prefer to reserve the term "mutations" in biology to changes of the DNA/RNA, that may or may lead to "substitutions" in proteins. Personally, I dislike the term "mutations" when speaking of proteins and rather see amino acid "substitutions" or "changes".

L240. RBD domain. Delete "domain".

L232-235. While increased affinity may increase infection efficiency, it may have the downside of reducing virus release (and potentially transmission)?

*Reviewer #2 (Recommendations for the authors):*

Additional data on the effects of variant mutations on ACE2 or cell binding using whole spikes or (pseudotyped) virus particles would add novelty.

*Reviewer #3 (Recommendations for the authors):*

(1) Did the authors use reducing or non-reducing SDS-PAGE in Figure S2 that could address whether disulfide-linked dimers are present?

(2) The "D" of "RBD" in the title already includes the meaning of "domain". Can the "Spike RBD domain" be replaced by "Spike receptor-binding domain" or simply "Spike RBD"?

(3) Page 5, line 82. Please change "20A/S:4.4K" to "20A/S.484K"

---

## [Author Response]

Essential revisions:No new experiments are required but numerous suggestions have been made by the 3 reviewers to correct or clarify the text.Reviewer #1 (Recommendations for the authors):Several countries have requested to not use country names in virus lineage descriptions, to avoid stigmatization. I would suggest to (maximally) once in the manuscript mention for each lineage where it was first detected, and afterwards refrain from using country names and country codes. In many tables and figures and legends and texts, these names recur (e.g. table 2, Figure S2 and elsewhere).

We have made the requested changes and added the new WHO variant nomenclature to the text, tables, figures, and figure legends.

Line 20. SARS-CoV-2 is not the second coronavirus that can induce a severe acute respiratory syndrome; all human coronaviruses can do that, and so can SARS-CoV and MERS-CoV.

We have corrected this.

Line 22. This is a matter of debate. I guess the authors only take infectious disease pandemics into account and do not consider the HIV pandemic to have unfolded in the past 100 years?

We appreciate that some might consider the HIV pandemic to have been worse that SARS-CoV-2 pandemic and have adjusted the text accordingly. We have also clarified that we are referring to infectious disease pandemics.

Line 61 and elsewhere. The authors mention "physiological temperature". It would be useful throughout the text to add the exact temperature used (37C) in statements like this. The temperature in the human airways is not the same everywhere (less than 37C in the upper airways). It is better to be exact.

We accept this and have amended the text.

On several occasions in the manuscript, the authors use "transmissivity" where they probably mean "transmissibility".

We have corrected this.

I prefer to reserve the term "mutations" in biology to changes of the DNA/RNA, that may or may lead to "substitutions" in proteins. Personally, I dislike the term "mutations" when speaking of proteins and rather see amino acid "substitutions" or "changes".

While we appreciate this point, we think is reasonable, in the interest of brevity, to use ‘mutations’ as shorthand for the phrase ‘mutations which results in the amino acid substitutions’. We think its meaning is clear.

L240. RBD domain. Delete "domain".

We have corrected this here and throughout the text, including in the title.

L232-235. While increased affinity may increase infection efficiency, it may have the downside of reducing virus release (and potentially transmission)?

While we agree that increased affinity could *decrease* transmission, e.g. via reducing viral release, this point is not obviously relevant in this study which is focused on variants whose spread suggests *increased* transmission.

Reviewer #3 (Recommendations for the authors):(1) Did the authors use reducing or non-reducing SDS-PAGE in Figure S2 that could address whether disulfide-linked dimers are present?

We used reducing SDS PAGE gels. We have added this to the Materials and methods and Figure 2—figure supplement 1 legend (previously Figure S2).

(2) The "D" of "RBD" in the title already includes the meaning of "domain". Can the "Spike RBD domain" be replaced by "Spike receptor-binding domain" or simply "Spike RBD"?

We have corrected this.

(3) Page 5, line 82. Please change "20A/S:4.4K" to "20A/S.484K"

We have corrected this.